# A homogeneous time-resolved fluorescence screen to identify SIRT2 deacetylase and defatty-acylase inhibitors

**Jie Yang**[1,2], **Joel Cassel**[3], **Brian C. Boyle**[1,2,4], **Daniel Oppong**[5], **Young-Hoon Ahn**[5], **Brian P. Weiser**[1,2] *

**1** Department of Molecular Biology, Rowan-Virtua School of Translational Biomedical Engineering & Sciences, Rowan University, Stratford, New Jersey, United States of America, **2** Department of Molecular Biology, Rowan-Virtua School of Osteopathic Medicine, Rowan University, Stratford, New Jersey, United States of America, **3** Molecular Screening & Protein Expression Facility, Wistar Institute, Philadelphia, Pennsylvania, United States of America, **4** Department of Biomedical Engineering, Rowan-Virtua School of Translational Biomedical Engineering & Sciences, Rowan University, Glassboro, New Jersey, United States of America, **5** Department of Chemistry, Drexel University, Philadelphia, Pennsylvania, United States of America

* weiser@rowan.edu

**Data Availability Statement:** High-throughput screening data for 9600 compounds tested in this work can be found in the Rowan Digital Works repository (https://rdw.rowan.edu/datasets/3).

## Abstract

Human sirtuin-2 (SIRT2) has emerged as an attractive drug target for a variety of diseases. The enzyme is a deacylase that can remove chemically different acyl modifications from protein lysine residues. Here, we developed a high-throughput screen based on a homogeneous time-resolved fluorescence (HTRF) binding assay to identify inhibitors of SIRT2's demyristoylase activity, which is uncommon among many ligands that only affect its deacetylase activity. From a test screen of 9600 compounds, we identified a small molecule that inhibited SIRT2's deacetylase activity ($IC_{50}$ = 7 μM) as well as its demyristoylase activity ($IC_{50}$ = 37 μM). The inhibitor was composed of two small fragments that independently inhibited SIRT2: a halogenated phenol fragment inhibited its deacetylase activity, and a tricyclic thiazolobenzimidazole fragment inhibited its demyristoylase activity. The high-throughput screen also detected multiple deacetylase-specific SIRT2 inhibitors.

## Introduction

The seven human sirtuin isoforms (SIRT1-SIRT7) perform a conserved $NAD^+$-dependent reaction to remove acyl modifications from lysine residues of proteins. The selectivity of sirtuins for acylated substrates in the cell is controlled on several levels. The isoforms have distinct subcellular localizations that control access to substrates [1], the enzymes recognize specific sequences surrounding acyl modifications [2], and some sirtuins are activated by co-factors such as DNA or nucleosomes [3–5]. Additionally, the sirtuin isoforms have strong preferences for different acyl modifications that occur on substrate lysines [6, 7]. For example, SIRT1, SIRT2, and SIRT3 are promiscuous deacetylases and defatty-acylases that can remove hydrophobic lysine modifications that are as long as 14 carbons (e.g., myristoylation), whereas

Other data from this work, including data presented in Figures, can be found on Zenodo (https://zenodo.org/records/11375633). All other relevant information is within the manuscript and its Supporting Information files.

**Funding:** Primary funding was from New Jersey Health Foundation grants to BPW (PC 61-21 and PC 1-22-13) (www.njhealthfoundation.org). The Weiser Lab was also supported by a National Institutes of Health grant (R01GM135152) (www.nih.gov). The funders did not play any role in the study design, data collection or analysis, decision to publish, or the preparation of the manuscript.

**Competing interests:** The authors have declared that no competing interests exist.

SIRT5 prefers to remove smaller polar groups from lysine residues such as succinyl or malonyl modifications [6–9].

The unique selectivity of sirtuins for different acyl modifications on proteins creates challenges and opportunities for drug discovery. Apart from creating isoform-specific inhibitors, there is interest in modulating select deacylase activities of sirtuins which likely have different roles in the cell [10–15]. A challenging issue related to sirtuin isoform 2 (SIRT2) specifically is that small molecules with good isoform selectivity and nM potency often inhibit its deacetylase activity without affecting its defatty-acylase/demyristoylase activity [16, 17]. Some exceptions to this include a peptide macrocycle inhibitor of SIRT2 and peptide-like, mechanism-based SIRT2 inhibitors that affect its demyristoylase activity; however, these peptide-like compounds generally lack drug-like qualities [18–22]. Newly identified molecules of the "SirReal" class have shown the most promise as SIRT2 demyristoylase inhibitors with potencies in the mid nM to low μM range [23, 24]. However, other small molecules that inhibit SIRT2's demyristoylase activity have lower selectivity or potency and include ascorbyl palmitate (IC$_{50}$ = ~8 to 23 μM) [25], 1-aminoanthracene (IC$_{50}$ = 21 μM) [12], "compound C" (IC$_{50}$ = 44 μM) [26], and suramin (IC$_{50}$ = 95 μM) [27]. It may be difficult to assess the therapeutic value of targeting different deacylase activities of SIRT2 without additional classes of defatty-acylase inhibitors.

New assays and screening methods are needed to increase the likelihood of identifying compounds that affect SIRT2's defatty-acylase activity along with its deacetylase activity. Previously reported high-throughput screens targeting SIRT2 focused on inhibiting its deacetylase activity [28–30] or identifying ligands that bind to a SIRT2—decanoyl-peptide complex [25]. New assays that are compatible with screening have been developed to detect inhibition of SIRT2's defatty-acylase activity [24, 26, 27, 31]. Assays developed to study other defatty-acylases including other sirtuin isoforms and classical histone deacetylases (HDACs) may also be adapted to identify SIRT2 inhibitors [32]. To add to this growing experimental toolbox, we report here the development of a high-throughput screen that detects SIRT2's interaction with a myristoylated substrate in a binding assay. We identified a small molecule from a test screen of 9600 compounds that competed with the myristoylated substrate for SIRT2 binding. The identified SIRT2 ligand inhibited the enzyme with potency that was comparable to most of the known SIRT2 demyristoylase inhibitors (IC$_{50}$ = 37 μM). We then investigated interactions of SIRT2 with chemical fragments from our identified inhibitor to understand the basis for deacetylase and demyristoylase inhibition. Our study provides new approaches to identify inhibitors of SIRT2's demyristoylase activity.

## Results and discussion

### Design and validation of a SIRT2—myristoyl-peptide HTRF binding assay

Previously, we reported a crystal structure of SIRT2 bound to a fluorescein-labeled, 13mer peptide whose sequence derived from histone H4 with lysine 16 myristoylated ("FAM-myristoyl-H4K16 peptide") (PDB code 8TGP) [33]. The structure showed that the enzyme binds the fluorescein-labeled peptide like a conventional myristoylated substrate (S1 Fig) [8, 34]. Only five amino acids from the substrate peptide were resolved in the crystal structure including the myristoylated lysine in the enzyme active site, two residues upstream of the modified lysine, and two residues downstream (S1 Fig). The N-terminal fluorescein, a PEG4 linker, and the first six residues of the peptide were not visible and were presumed to be hanging off the protein freely in the crystal lattice. We confirmed that SIRT2 can remove the myristoyl modification from FAM-myristoyl-H4K16 peptide [33], and we confirmed with a gel shift assay [33] and with isothermal titration calorimetry (ITC) (S2 Fig) that SIRT2 binds the fluorescein-labeled peptide with nearly the same affinity as an identical, unlabeled myristoyl-H4K16

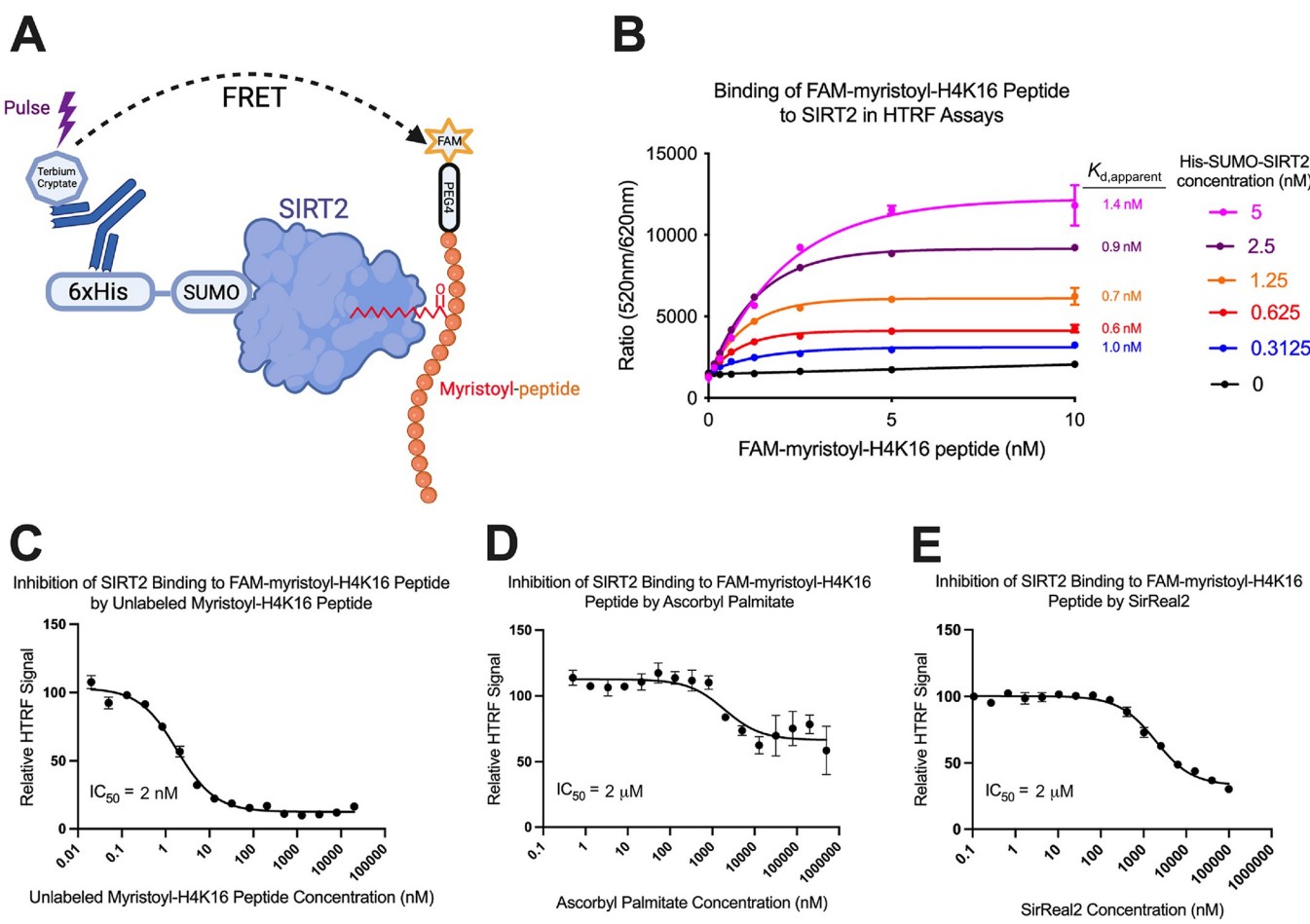

**Fig 1. Design and validation of a SIRT2—myristoyl-peptide HTRF binding assay.** (A) Schematic of our HTRF binding assay where His tagged-SIRT2 interacts with a terbium cryptate-labeled antibody and a fluorescein-labeled myristoyl-peptide. Displacement of the peptide from SIRT2 by a competitor then reduces FRET efficiency after terbium cryptate excitation. (B) Fluorescence intensity in HTRF binding assays using different concentrations of SIRT2 and FAM-myristoyl-H4K16 peptide (excitation/emission wavelengths = 330 nm/520 nm, normalized by emission at 620 nm) (see Materials and Methods). (C) Displacement of FAM-myristoyl-H4K16 peptide from SIRT2 in the HTRF binding assay by an identical peptide lacking the FAM-PEG4 group. (D) Displacement of FAM-myristoyl-H4K16 peptide from SIRT2 in the HTRF binding assay by the SIRT2 deacetylase and defatty-acylase inhibitor ascorbyl palmitate. (E) Displacement of FAM-myristoyl-H4K16 peptide from SIRT2 in the HTRF binding assay by the SIRT2 deacetylase inhibitor SirReal2.

peptide that lacks the FAM-PEG4 moiety. We concluded that FAM-myristoyl-H4K16 peptide accurately reports on SIRT2's interaction with myristoylated substrate in binding and activity assays, that the FAM-PEG4 moiety does not contribute to or affect peptide binding, and that the peptide was an ideal substrate to use in screening experiments that aimed to identify SIRT2 demyristoylase inhibitors.

Here, we designed a homogeneous time-resolved fluorescence (HTRF) binding assay to detect the interaction of SIRT2's catalytic domain with FAM-myristoyl-H4K16 peptide (Fig 1A). In summary, purified SIRT2 containing a 6xHis-SUMO tag on its N-terminus was complexed with a terbium cryptate-labeled anti-His tag antibody (Fig 1A). Excitation of terbium cryptate ($\lambda$ = 330 nm) produced a fluorescence emission that could serve as a FRET donor for the fluorescein label on FAM-myristoyl-H4K16 peptide, which has peak excitation/emission wavelengths of 495 nm/520 nm. Therefore, FRET could occur when the peptide was bound to SIRT2 and held in proximity to the antibody, as measured by the emission of fluorescein at 520 nm following terbium cryptate excitation at 330 nm. Additionally, displacement of

FAM-myristoyl-H4K16 peptide from SIRT2 by a competitor then reduces FRET efficiency and the emission of fluorescein.

The interaction of SIRT2 with FAM-myristoyl-H4K16 peptide was measured using the HTRF assay in 384-well plates. The fluorescence of 10 nM FAM-myristoyl-H4K16 peptide increased ~6-fold when equilibrated with 5 nM SIRT2 and 0.2 nM of terbium cryptate-labeled antibody (excitation/emission wavelengths = 330 nm/520 nm) (Fig 1B). The sub-stoichiometric concentration of antibody compared to SIRT2 minimized the presence of free antibody in solution and any non-specific/SIRT2-independent FRET that might occur between the terbium cryptate label and the peptide. We confirmed that the fluorescence increase resulted from FRET because, without the terbium cryptate-labeled antibody, the fluorescence of FAM-myristoyl-H4K16 peptide did not significantly change when bound to SIRT2 (S3 Fig). Using a range of protein and peptide concentrations (0–10 nM), we measured a $K_{d,apparent}$ of ~1 nM for the interaction of SIRT2 with FAM-myristoyl-H4K16 peptide (Fig 1B).

Our final conditions for HTRF-based assays were 10 μl/well with 3 nM SIRT2, 0.2 nM terbium cryptate-labeled antibody, and 3 nM FAM-myristoyl-H4K16 peptide. We determined a Z' of 0.78 under these screening conditions with no effect of DMSO at concentrations up to 1% (S4 Fig). To further validate the HTRF binding assay, we competed FAM-myristoyl-H4K16 peptide from SIRT2 with an unlabeled myristoyl-H4K16 peptide (Fig 1C). The unlabeled peptide efficiently and competitively displaced the FAM-labeled peptide from SIRT2 and reduced the FRET signal by ~90% (Fig 1C). The $IC_{50}$ of 2 nM for unlabeled myristoyl-H4K16 binding agreed with the $K_{d,apparent}$ of ~1 nM that we measured for the interaction of FAM-myristoyl-H4K16 peptide with SIRT2. Thus, we detected equivalent binding to SIRT2 for the labeled and unlabeled myristoyl-peptides. We also competed FAM-myristoyl-H4K16 peptide from the enzyme with known SIRT2 inhibitors, ascorbyl palmitate and SirReal2 [25, 28]. Ascorbyl palmitate is a dual deacetylase/demyristoylase inhibitor that presented as a partial inhibitor of myristoyl-peptide binding with a relative $IC_{50}$ of 2 μM (Fig 1D), which was in range of its published $K_d$ for SIRT2 ($K_d$ = 3 to 5 μM) [25]. Ascorbyl palmitate reduced the HTRF FRET signal by a maximum of only ~40% at the highest ligand concentrations (Fig 1D). SirReal2 also partially displaced FAM-myristoyl-H4K16 peptide by ~70% with a relative $IC_{50}$ of 2 μM (Fig 1E). This was notable because SirReal2 is not a competitive demyristoylase inhibitor and selectively inhibits SIRT2's deacetylase activity [16]. Thus, this screen has the potential to identify dual deacetylase/demyristoylase inhibitors of SIRT2 as well as ligands that affect its deacetylase activity either selectively or with greater potency than its demyristoylase activity.

## Library screening to identify SIRT2 ligands

9600 chemically diverse, lead-like compounds were tested at a single concentration of 10 μM for their ability to interact with SIRT2 by displacing FAM-myristoyl-H4K16 peptide from the enzyme's active site (Fig 2A). 63 of the compounds reduced binding of the fluorescein-labeled peptide by >40% in the initial screen. These 63 compounds were then examined with the same HTRF assay in full dose-response experiments to confirm binding (S5 Fig). Only 17% of the initial hits (11 of the 63 compounds) were confirmed to reduce FAM-myristoyl-H4K16 peptide binding by >40% (S5 Fig); additional plate replicates may be needed to reduce the false positive rate in future screens. In the full dose-response assays, we found that 9 compounds displaced FAM-myristoyl-H4K16 peptide from SIRT2 with $IC_{50}$ values less than 10 μM (Fig 2A) (see S6 Fig for the chemical structures of all 9 compounds). These 9 compounds were tested for their ability to inhibit SIRT2 deacetylase and demyristoylase activities using unlabeled H4K16 peptides containing an acetyl or myristoyl modification as the substrates. All 9 compounds at a concentration of 10 μM moderately inhibited SIRT2's deacetylase

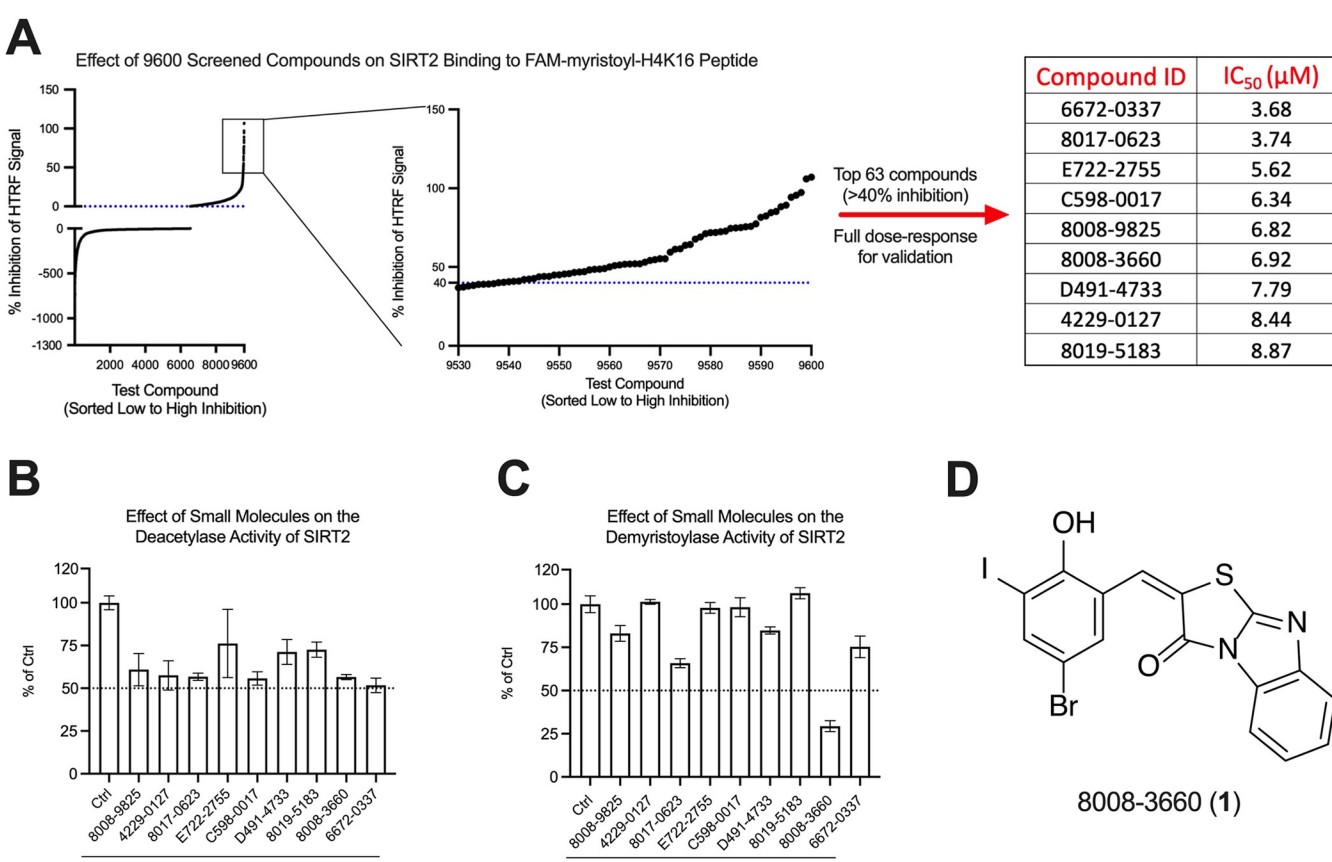

**Fig 2. High-throughput screen of 9600 compounds for SIRT2 binding in HTRF assays and the identification of 8008–3660 (1) as a SIRT2 deacylase inhibitor.** (A) 9600 compounds at a single concentration of 10 μM were screened for their ability to displace FAM-myristoyl-H4K16 peptide from SIRT2. The percent inhibition of peptide binding caused by all 9600 compounds is shown in the left panel. The middle panel shows the data for test compounds that inhibited SIRT2 binding to FAM-myristoyl-H4K16 peptide by >40%. These 63 compounds were used in full dose-response binding assays in HTRF format to confirm binding and to determine an IC50 for peptide displacement (S5 Fig). The 9 ligands in the table in the right of panel A inhibited FAM-myristoyl-H4K16 peptide binding to SIRT2 with IC50 values less than 10 μM. See S6 Fig for the chemical structures of all 9 compounds. (B) Effect of 10 μM compound on the deacetylase activity of SIRT2. The enzyme activity in the presence of compound was compared to a control SIRT2 assay (Ctrl) where the enzyme was treated with DMSO alone. (C) Effect of 50 μM compound on the demyristoylase activity of SIRT2 compared to a control SIRT2 assay (Ctrl) where the enzyme was treated with DMSO alone. Assays in panels B and C contained 0.7% DMSO from dissolving and diluting ligands. (D) Chemical structure of the identified SIRT2 ligand 8008–3660 (1).

activity (Fig 2B). We tested 50 μM of each compound in demyristoylase assays because this activity has historically been more difficult to affect pharmacologically [16]. While several compounds also inhibited SIRT2's demyristoylase activity, the ligand "8008–3660" (**1**) was considerably most effective (Fig 2C and 2D). Thus, we continued to characterize the properties of **1** and its chemotype.

## Inhibition of SIRT2 deacetylase and defatty-acylase activities with μM potency

A secondary binding assay was performed to independently confirm binding of **1** to SIRT2. We previously characterized the interaction of SIRT2 with a Cy3-labeled myristoyl-H4K16 peptide, which binds SIRT2 identical to unlabeled myristoyl peptide, but has enhanced fluorescence when associated with protein [25, 33]. Displacement of Cy3-myristoyl-H4K16 peptide from SIRT2 by **1** produced an IC50 of 16 μM (Fig 3A), which could be used to calculate a $K_d$ of

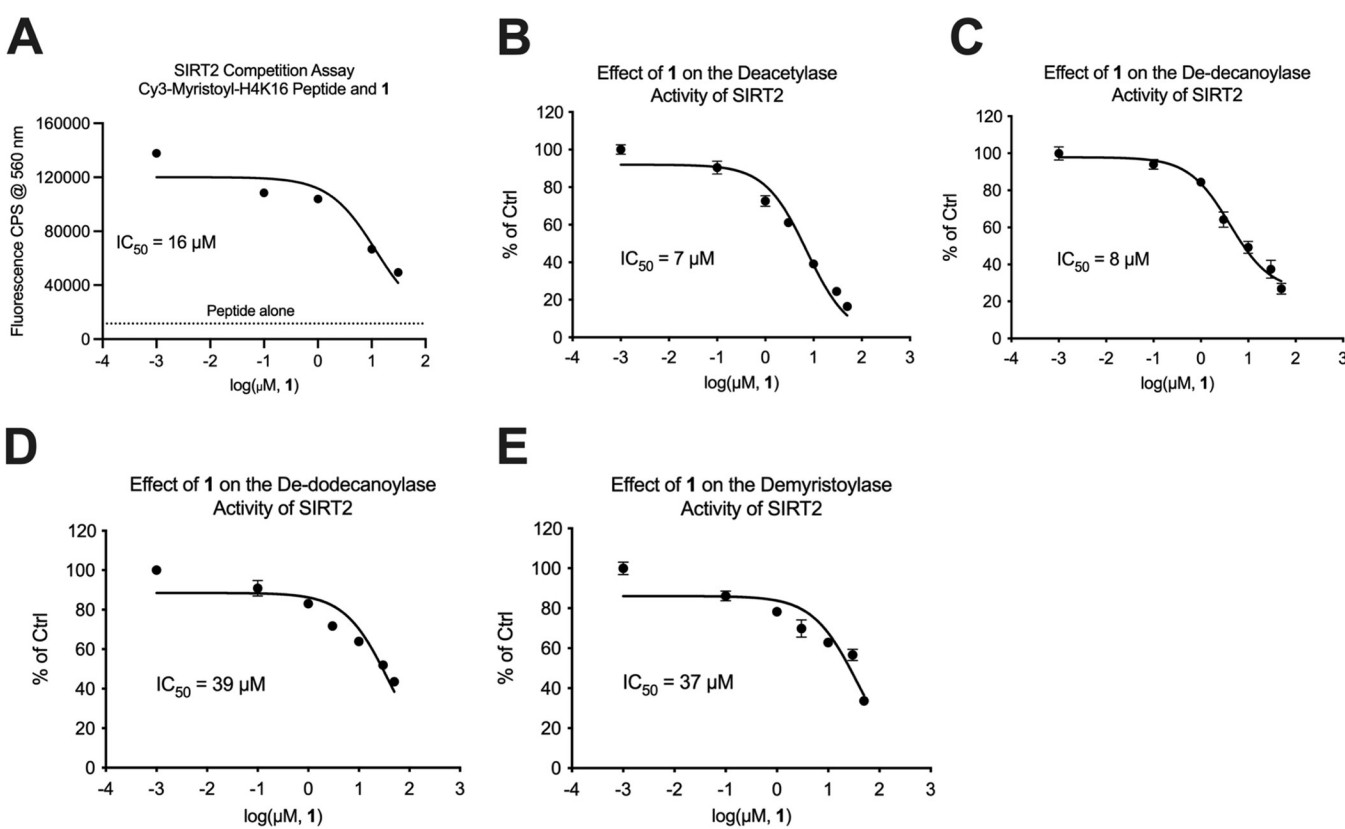

**Fig 3. SIRT2 binding and inhibition of SIRT2 deacylase activities by 1.** (A) Displacement of Cy3-myristoyl-H4K16 peptide from SIRT2 by **1** as measured by a reduction in Cy3 fluorescence. The $IC_{50}$ value of 16 µM was used to calculate a $K_d$ of 5 µM for the interaction of **1** with SIRT2 (see Materials and Methods). (B) **1** inhibited SIRT2 deacetylase activity with an $IC_{50}$ of 7 µM. (C) **1** inhibited SIRT2 de-decanoylase activity with an $IC_{50}$ of 8 µM. (D) **1** inhibited SIRT2 de-dodecanoylase activity with an $IC_{50}$ of 39 µM. (E) **1** inhibited SIRT2 demyristoylase activity with an $IC_{50}$ of 37 µM. See Materials and Methods for assay conditions.

5 µM for the interaction of **1** with SIRT2 (see Materials and Methods) [12, 25, 35, 36]. Full-dose response activity assays were conducted with unlabeled H4K16 peptides to determine $IC_{50}$ values for inhibiting SIRT2 deacylase activities. For **1**, the $IC_{50}$ values for inhibiting deacetylase and de-decanoylase activities (7–8 µM) were five-fold more potent than the $IC_{50}$ values for inhibiting de-dodecanoylase and demyristoylase activities (37–39 µM) (Fig 3B–3E). This was interesting because we consider decanoylation to be a long fatty acyl modification with relatively high affinity for SIRT2 compared to acetylation [12, 33], and **1** preferentially inhibited deacylase activity at a very specific acyl chain length. To gauge the selectivity of **1** for SIRT2 compared to other human sirtuins, we also determined that **1** inhibited SIRT1 deacetylase activity with an $IC_{50}$ of 32 µM while having no effect on the demyristoylase activity of SIRT6 (S7 Fig).

We then examined whether **1** could inhibit SIRT2 in cells. Inhibition of SIRT2's deacetylase activity in cells can result in elevated levels of acetylated alpha-tubulin, which is an *in vivo* substrate of SIRT2 [16, 18, 19, 28, 37]. As measured with immunofluorescence, treatment of cells with 100 µM of **1** significantly increased the level of acetylated alpha-tubulin (Fig 4A and 4B). We performed two control experiments to further substantiate that the change in acetylated alpha-tubulin resulted from SIRT2 inhibition by **1** because this readout indirectly reports on SIRT2 activity in cells, as opposed to directly measuring engagement of the ligand with the enzyme [38]. First, we treated cells with 25 µM of the SIRT2 deacetylase inhibitor SirReal2

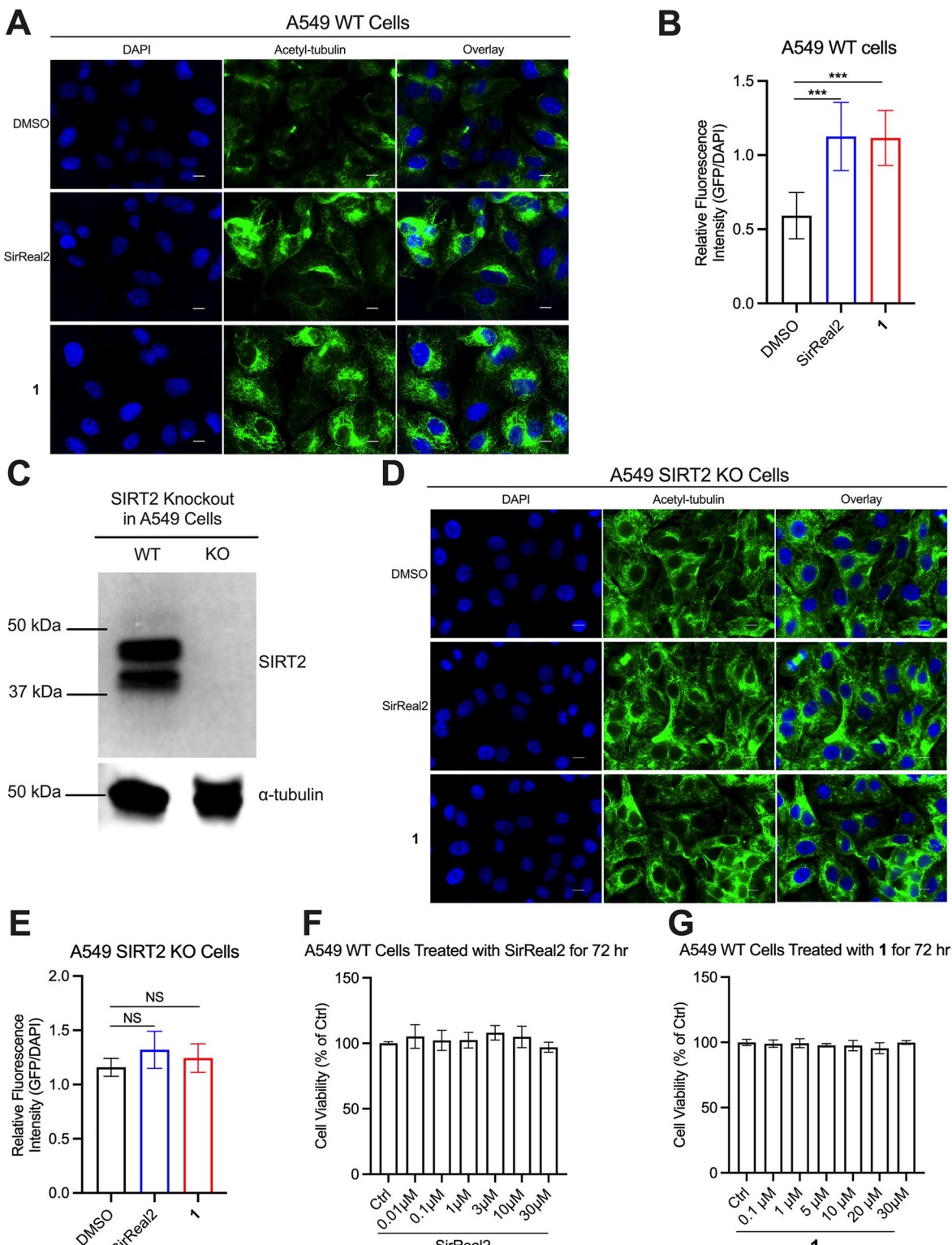

**Fig 4. Inhibition of SIRT2 by 1 in cells.** (A) Immunofluorescence images of acetylated alpha-tubulin in A549 cells showing enhanced fluorescence in cells treated with 25 μM of SirReal2 (positive control) or 100 μM of **1**. The media for the control contained 2% DMSO, and the scale bar is 10 μm. (B) Comparison of GFP fluorescence intensities from acetylated alpha-tubulin staining from the images in panel A. GFP fluorescence intensities from individual cells were measured and normalized to the DAPI intensities of the same cells, and the intensities were compared with a one-way ANOVA followed by Dunnett's multiple comparison test (***p < 0.001). (C) Western blot

showing knockout of SIRT2 in A549 cells using CRISPR/Cas9. The alpha-tubulin blot, which was performed on a separate membrane, was used as a loading control. (D) Immunofluorescence images of A549 SIRT2-KO cells showing increased acetylated alpha-tubulin levels in cells without the protein, and the ligands SirReal2 and **1** had no effect on the levels of acetylated alpha-tubulin in SIRT2-KO cells. Experiments were performed identical to panel A. (E) Comparison of GFP fluorescence intensities from acetylated alpha-tubulin staining from the images in panel D. These were quantified and compared as described above for panel B. (F) SirReal2 treatment for 72 hours did not cause toxicity in A549 cells, as measured with an MTT assay. The control (Ctrl) cells were treated with 0.7% DMSO to be consistent with the drug treated cells. (G) Treatment of A549 cells with **1** for 72 hours did not cause toxicity, which was determined with an MTT assay as in panel F.

[28], which increased acetylated alpha-tubulin to similar levels as **1** (Fig 4A and 4B). Secondly, we generated A549 SIRT2-KO cells (Fig 4C), which also had an elevated level of acetylated alpha-tubulin that was unchanged by the SIRT2 inhibitors (Fig 4D and 4E).

Finally, as discussed elsewhere [39, 40], SIRT2 regulates diverse processes in cancer cells depending on the tissue type and tumor stage, causing the enzyme to be called both a tumor suppressor and oncogene, and its inhibition has varying effects on cellular growth rates. In our case, we found that the SIRT2 inhibitors SirReal2 and **1** had no effect on the viability of A549 cells as measured with an MTT assay (Fig 4F and 4G).

## Exploration of lead fragments and the phenolic chemotype

The lead compound **1** essentially contained two small molecule fragments that were linked together by a rotatable bond (Fig 5). We obtained the two fragments 4-bromo-2-iodophenol (**2**) and the tricyclic thiazolobenzimidazole compound (**15**) (Fig 5). We determined that **2** interacted with SIRT2 with a $K_d$ of 68 μM, inhibited SIRT2 deacetylase activity with an $IC_{50}$ of 9 μM, but had no effect on SIRT2 demyristoylase activity at concentrations as high as 200 μM (Fig 5 and S8 Fig). In contrast, **15** was not effective as a deacetylase inhibitor, but was a weak demyristoylase inhibitor ($IC_{50}$ = 117 μM) (Fig 5 and S8 Fig). The interaction of **15** with SIRT2 was too weak to measure a $K_d$ (S8 Fig). It is plausible that fragment **2** was responsible for the efficacy of **1** as a SIRT2 deacetylase inhibitor, and that fragment **15** was responsible for its efficacy as a demyristoylase inhibitor. The finding that fragments **2** and **15** independently inhibit SIRT2 explains why their potency increases when they are joined together in **1**.

Coincidentally, we noted that fragment **2** strongly resembled propofol (**3**), which is a clinically-used general anesthetic that we previously characterized as a weak SIRT2 inhibitor (Fig 5) [12, 41]. **3** binds SIRT2 with a $K_d$ of 25 μM and inhibits deacetylase and demyristoylase activities with $IC_{50}$ values between 140–195 μM (Fig 5) [12]. We obtained several other alkylphenols that are FDA-approved as pharmaceutical and food additives (**4**–**6**). These also interacted with SIRT2, but with weaker affinity than **1**, **2**, and **3** (Fig 5 and S9 Fig). The original alkylphenol that was discovered to bind SIRT2 was an experimental compound called *meta*-azi-propofol (**7**) that acts as a photolabel by virtue of its trifluoromethyl diazirine group [41–44]. We found that a commercially-available photolabel analog lacking the hydroxyl (**8**) was a weak deacetylase inhibitor similar to propofol ($IC_{50}$ = 114 μM) (Fig 5). We obtained several more halogenated phenols with various substitutions at different ring positions (**9**–**14**), but none affected SIRT2 activity at concentrations as high as 200 μM. We were surprised by the relative potency of **2** in deacetylase assays compared to analogs **9**–**14** considering the similarity between their size, volume, and polarity (S1 Table). These compounds are lipophilic and uncharged at neutral pH, and they should bind similarly sized hydrophobic sites [45–47]. However, it is noteworthy that **2** has fewer halogens than the ineffective ligands. As a final control on the stability of **2**, we used intact protein mass spectrometry to confirm that **2** does not irreversibly attach to SIRT2 or form any type of covalent adduct on the protein after incubation at 37°C (S10 Fig).

| | | SIRT2 $K_d$ (μM) | Deacetylase IC$_{50}$ (μM) | Demyristoylase IC$_{50}$ (μM) | | | SIRT2 $K_d$ (μM) | Deacetylase IC$_{50}$ (μM) | Demyristoylase IC$_{50}$ (μM) |
|---|---|---|---|---|---|---|---|---|---|
| **1** | | 5 | 7 | 37 | **9** | | N.D. | no effect | no effect |
| **2** | | 68 | 9 | no effect | **10** | | N.D. | no effect | no effect |
| **3** | | 25* | 140* | 195* | **11** | | N.D. | no effect | no effect |
| **4** | | 116 | N.D. | N.D. | **12** | | N.D. | no effect | no effect |
| **5** | | 117 | N.D. | N.D. | **13** | | N.D. | no effect | no effect |
| **6** | | 73 | N.D. | N.D. | **14** | | N.D. | no effect | no effect |
| **7** | | N.D. | N.D. | N.D. | **15** | | N.D. | no effect | 117 |
| **8** | | N.D. | 114 | no effect | | | | | |

**Fig 5. The ability of 15 small molecules to bind SIRT2 or affect its deacetylase or demyristoylase activities.** "N.D." means not determined, and "no effect" indicates that 200 μM compound did not significantly affect SIRT2 activity. The $K_d$ value for **1** was determined from a Cy3-myristoyl-H4K16 peptide binding competition assay. The $K_d$ values for **2**, **4**, **5**, and **6** were determined from a 1-aminoanthracene binding competition assay. *All data for **3** was previously published [12]. See Materials and Methods for experimental details and the names of the chemicals.

## Substrate-dependent interactions between fragment 2 and SIRT2

We previously determined that **3** binds within SIRT2's hydrophobic tunnel that also accommodates its substrate acyl chains [12]. A unique characteristic was that different acyl

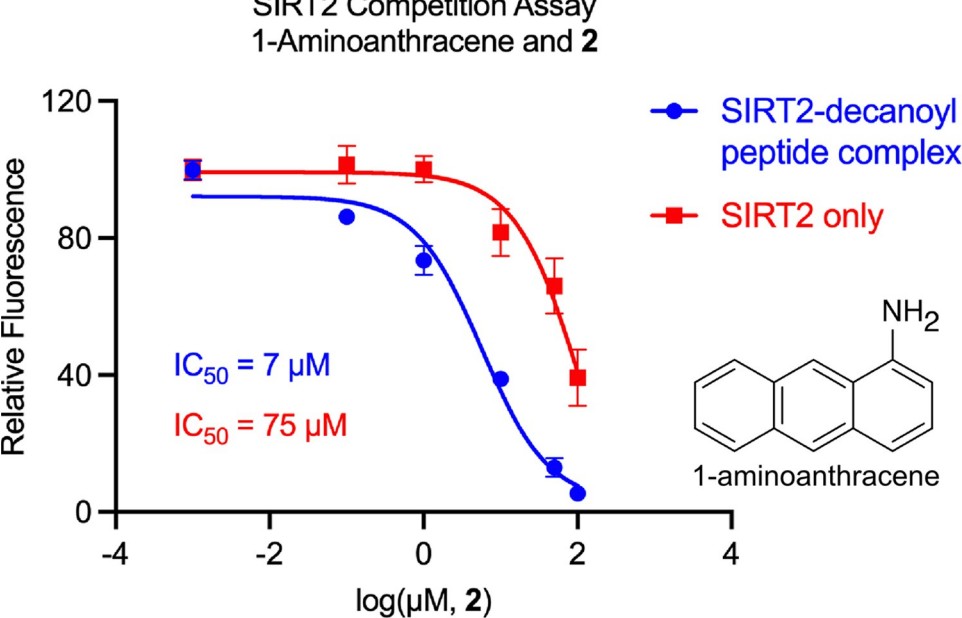

**Fig 6. Enhanced affinity of 2 for the SIRT2—decanoyl-peptide complex compared to SIRT2 alone.** 2 was used to displace the fluorescent ligand 1-aminoanthracene from SIRT2's hydrophobic tunnel which resulted in a reduction of 1-aminoanthracene fluorescence. The concentration of 1-aminoanthracene was 100 nM, the SIRT2 concentration was 4 μM, and when added, the concentration of decanoyl-H4K16 peptide was 10 μM. The IC$_{50}$ values were used to calculate $K_d$ values of 68 μM for the interaction of 2 with SIRT2 and 3 μM for the interaction of 2 with the SIRT2—decanoyl-peptide complex [12]. The $K_d$ of 68 μM for its SIRT2 interaction was also reported in Fig 5.

modifications affected the interaction of **3** with SIRT2 in different ways. Myristoyl groups occupy the entire hydrophobic tunnel and bind competitively with **3**; however, the shorter decanoyl-lysine modification and **3** can simultaneously occupy SIRT2's hydrophobic tunnel [12]. This stabilizes the binding site of **3** on SIRT2; **3** binds the SIRT2—decanoyl peptide complex with a $K_d$ of 7 μM compared to a $K_d$ of 25 μM for SIRT2 alone [12]. Affinity for SIRT2 and the SIRT2—decanoyl peptide complex can be measured in competition assays using the fluorescent ligand 1-aminoanthracene, which binds SIRT2 at the same site as **3**; 1-aminoanthracene binds the SIRT2—decanoyl peptide complex with a $K_d$ of 4 μM compared to a $K_d$ of 37 μM for SIRT2 alone [12]. Here, we determined that fragment **2** also preferred to bind the SIRT2—decanoyl peptide complex. Binding competition assays with 1-aminoanthracene determined a $K_d$ of 3 μM for the interaction of **2** with the SIRT2—decanoyl-peptide complex, which was 23-fold stronger than its affinity for SIRT2 alone ($K_d$ = 68 μM) (Fig 6). Although we attempted to measure the affinity of **1** for the SIRT2—decanoyl peptide complex, this could not be determined because **1** absorbed light in the same wavelength range as 1-aminoanthracene.

The experiment in Fig 6 positions the binding site of fragment **2** at the end of the hydrophobic tunnel near residues that interact with **3** and 1-aminoanthracene, including Tyr139, Phe190, and Leu206 [12, 41]. This is an allosteric site relative to a bound acetyl-lysine substrate, but a competitive site relative to a bound myristoyl-lysine. In the context of **1** binding to SIRT2, whether fragment **15** would orient inside the hydrophobic tunnel or in an adjacent allosteric site to inhibit demyristoylase activity is not yet clear.

## Conclusion

In this report, we developed and validated a high-throughput screen using HTRF to identify dual SIRT2 deacetylase/demyristoylase inhibitors. The hit compound **1** (8008–3660) had

similar potency as many other reported small molecule inhibitors of SIRT2's demyristoylase activity. **1** is composed of two small fragments that independently inhibited SIRT2's deacetylase or demyristoylase activity. This hints that the different deacylase activities of SIRT2 could be independently targeted by optimizing individual fragments that, when tethered together, would act as dual deacetylase/demyristoylase inhibitors. Finally, the phenolic fragment **2** (4-bromo-2-iodophenol) that we discovered as a SIRT2 deacetylase inhibitor is of the same chemotype as **3** (propofol), which is a clinically-used general anesthetic. Our findings provide new approaches to inhibit the different deacylase activities of SIRT2.

## Materials and methods

### Protein, peptide, and chemical materials

For high-throughput screening, we used a human SIRT2 catalytic domain protein (amino acids 34–356) that contained a 6xHis-SUMO tag on its N-terminus [19, 25]. We removed the 6xHis-SUMO tag and chromatographically purified the SIRT2 catalytic domain for all other binding and activity assays. The proteins were expressed and purified as previously described [12, 19, 33]. The synthetic peptides we used contained a 13 residue histone H4 sequence with an acyl modification on lysine 16 (H4K16). All peptides were obtained from New England Peptide and were described previously [12, 25, 33]. All high-throughput screening compounds including 8008–3660 (**1**) were purchased from Chemical Diversity/ChemDiv. Other chemicals at a purity of at least 97% were purchased from the following vendors: 4-bromo-2-iodophenol (**2**), Ambeed; 2-6-diisopropylphenol (propofol) (**3**), Alfa Aesar; 2-isopropyl-5-methylphenol (thymol) (**4**), TCI America; 5-isopropyl-2-methylphenol (carvacrol) (**5**), TCI America; 5-methyl-2-pentylphenol (amylmetacresol) (**6**), MilliporeSigma; 3-phenyl-3-(trifluoromethyl)-3H-diazirine (**8**), TCI America; 4-bromo-2-fluoro-6-iodophenol (**9**), AOBChem USA; 4-bromo-2,6-diiodophenol (**10**), Ambeed; 4-bromo-2-(trifluoromethyl)phenol (**11**), TCI America; 4-bromo-2,6-difluorophenol (**12**), Matrix Scientific; 4-butoxy-2,6-difluorophenol (**13**), Apollo Scientific; 2-fluoro-5-(trifluoromethyl)phenol (**14**), Matrix Scientific; [1,3] thiazolo[3,2-a]benzimidazole-3(2H)-one (**15**), Matrix Scientific.

### SIRT2 HTRF binding assay and high-throughput screen

HTRF binding measurements were made at room temperature in white, low volume 384-well plates using a ClarioStar plate reader. The assay volume was 10 µl, and the buffer contained 10 mM Hepes-NaOH, pH 7.4, 150 mM NaCl, 5 mM DTT, and 0.005% Tween-20. Each well contained a final concentration of 0.2 nM anti-6xHis terbium cryptate-labeled antibody (Cisbio catalog #61HI2TL) that served as the FRET donor. Initial assays that tested binding between protein and peptide in HTRF format had varying amounts of 6xHis-SUMO-SIRT2 and FAM-myristoyl-H4K16 peptide in the wells (0–10 nM each) (Fig 1B). Hyperbolic binding curves were fit to this data using the equation

$$y = Y_{min} + (Y_{max} - Y_{min})*(1 - \exp(-k*x)) \tag{1}$$

where $Y_{min}$ was the estimated fluorescence at 0 µM FAM-myristoyl-H4K16 peptide, $Y_{max}$ was the theoretical fluorescence at infinite concentrations of FAM-myristoyl-H4K16 peptide, and $k$ was a rate constant. The $K_{d,apparent}$ for each curve was determined as the concentration of FAM-myristoyl-H4K16 peptide that yielded fluorescence at the halfway point between the $Y_{min}$ and $Y_{max}$. We report binding affinities as $K_{d,apparent}$ values instead of true $K_d$ values because SIRT2 was not stoichiometrically labeled with the terbium cryptate antibody, and neither SIRT2 nor FAM-myristoyl-H4K16 peptide were concentration-limited in the assay.

For high-throughput screening, each well contained a final concentration of 3 nM 6xHis-SUMO-SIRT2 and 3 nM FAM-myristoyl-H4K16 peptide along with 1% DMSO, 10 μM test compound, and 0.2 nM anti-6xHis terbium cryptate-labeled antibody. Control wells omitted 6xHis-SUMO-SIRT2 or test compound while maintaining 1% DMSO. Test compounds were from a custom library from Chemical Diversity/ChemDiv of compounds containing lead-like properties focused on chemical diversity, solubility, topology, and Fsp3 character among other characteristics.

HTRF measurements were collected in the following manner. All components except the FAM-myristoyl-H4K16 peptide were equilibrated in a 5 μl volume for 15–30 min before the addition of 5 μl diluted peptide. 1 hr after adding the peptide, terbium cryptate was excited in the plate reader using 330 nm followed by emission measurements at 520 nm (for fluorescein FRET) and 620 nm (for an internal control). The delay for emission recording after the excitation flash was 60 μs or 50 μs for 520 nm and 620 nm, respectively, and the integration time for emission collection was 400 μs or 200 μs. 100 excitation flashes were made per well. For data processing, FRET emission measurements at 520 nm were divided by the emission of terbium cryptate at 620 nm, which normalized the data by controlling for the amount of terbium cryptate-labeled antibody in the well. Our quantification of FRET fluorescence in HTRF assays therefore referred to the normalized fluorescence ratio (520 nm/620 nm).

The displacement of FAM-myristoyl-H4K16 peptide from SIRT2 during the high-throughput screen was quantified by calculating the percent of binding inhibition using the equation

$$100 - \left(\frac{C_{fluor} - N_{fluor}}{D_{fluor} - N_{fluor}}\right)*100 \tag{2}$$

where $C_{fluor}$ was the FRET fluorescence in the presence of test compound, SIRT2, and FAM-myristoyl-H4K16 peptide, $N_{fluor}$ was the FRET fluorescence with FAM-myristoyl-H4K16 peptide and 1% DMSO (no SIRT2), and $D_{fluor}$ was the FRET fluorescence in the presence of 1% DMSO, SIRT2, and FAM-myristoyl-H4K16 peptide. Note that a negative percent inhibition occurs when the small molecule enhances FRET fluorescence. The Z' we report was calculated using the equation

$$1 - \frac{3(\sigma_N + \sigma_D)}{|\mu_N + \mu_D|} \tag{3}$$

where $\sigma_N$ was the standard deviation of FRET fluorescence with FAM-myristoyl-H4K16 peptide and 1% DMSO (no SIRT2), $\sigma_D$ was the standard deviation of FRET fluorescence in the presence of 1% DMSO, SIRT2, and FAM-myristoyl-H4K16 peptide, $\mu_N$ was the mean FRET fluorescence with FAM-myristoyl-H4K16 peptide and 1% DMSO, and $\mu_D$ was the mean FRET fluorescence with 1% DMSO, SIRT2, and FAM-myristoyl-H4K16 peptide [48].

## SIRT2 activity assays

The enzymatic activity of SIRT2 was measured using a MALDI-MS—based protocol reported previously [25]. Reactions were performed at 37°C for 2.5 min in PBS containing 1 mM DTT and 1 mM of $NAD^+$. Acylated H4K16 peptides were used as the substrates at the following concentrations: acetyl-H4K16, 4 μM; decanoyl-H4K16, 0.2 μM; dodecanoyl-H4K16, 0.5 μM; myristoyl-H4K16, 0.5 μM. These concentrations correspond to the enzyme's $K_m$ for each substrate under saturating $NAD^+$ conditions [12]. The enzyme concentration in each reaction was chosen such that 20% or less of the substrate was processed by the enzyme when the reactions were quenched [12, 25]. In dose-response assays showing a reduction in SIRT2 activity by

increasing amounts of ligand, the data was fit with standard sigmoidal dose-response curves as we previously described [12, 36]. We report absolute $IC_{50}$ values from the curves which reflect ligand concentrations that reduce the activity of the enzyme by 50% compared to control (no ligand) samples. Assays measuring the effects of ligand on SIRT1 deacetylase and SIRT6 demyristoylase activities were performed similarly, as previously described [25]. The concentration of substrate peptide in those assays were 1 μM acetyl-H4K16 or 16 μM myristoyl-H4K16, which also approximated the $K_m$ values for SIRT1 or SIRT6 processing the peptides under saturating $NAD^+$ concentrations [25].

## SIRT2 binding competition assays using Cy3-myristoyl-H4K16 peptide or 1-aminoanthracene

Competition assays where Cy3-myristoyl-H4K16 peptide was displaced from SIRT2 by **1** were performed as previously described [25]. Assays were performed in PBS with 1 mM DTT at 23°C in a quartz microcuvette (160 μl assay volume). Briefly, the fluorescence intensity of 50 nM Cy3-myristoyl-H4K16 peptide was measured in the presence of 2.5 μM SIRT2 and increasing amounts of **1** using excitation/emission wavelengths of 535 nm/560 nm. The final DMSO concentration in each measurement was 0.6%. As previously described [25], displacement of Cy3-myristoyl-H4K16 peptide from SIRT2 by a competitive ligand results in a reduction of its fluorescence.

Competition assays where 1-aminoanthracene was displaced from SIRT2 by **2**, **4**, **5**, and **6** were performed as previously reported [12] where we described its displacement from SIRT2 by **3**. The assays were also cuvette-based and performed at 23°C in PBS with 1 mM DTT, and the final DMSO concentration was 0.6%. Briefly, the fluorescence intensity of 100 nM 1-aminoanthracene was measured in the presence of 4 μM SIRT2 and increasing amounts of ligand using excitation/emission wavelengths of 390 nm/520 nm. As previously described [12, 25], displacement of 1-aminoanthracene from SIRT2 by a competitor results in a reduction of its fluorescence.

The data from both competition assays show a reduction in probe fluorescence at increasing concentrations of competitor ligands. This data could be fit with standard sigmoidal dose-response curves [12, 36]. We determined and report absolute $IC_{50}$ values from the curves which reflect ligand concentrations that reduce the fluorescence by 50% compared to control (no ligand) samples. $IC_{50}$ values from binding competition assays were used to calculate $K_d$ values for the interaction of SIRT2 with the competitor ligands using the equation

$$K_i = \frac{IC_{50}}{\left(\frac{L_{50}}{K_d}\right) + \left(\frac{P_0}{K_d}\right) + 1} \quad (4)$$

where $K_i$ was the calculated $K_d$ for the interaction of competitor ligand with SIRT2, $IC_{50}$ was determined from the binding competition assay for each competitor, $L_{50}$ was the concentration of free fluorescent probe at 50% inhibition (either Cy3-myristoyl-H4K16 peptide or 1-aminoanthracene), $K_d$ was the dissociation constant for the SIRT2—Cy3-myristoyl-H4K16 peptide complex (1 μM) or the dissociation constant for the SIRT2–1-aminoanthracene complex (37 μM) depending on the assay, and $P_0$ was the concentration of free SIRT2 at 0% inhibition [12, 25, 35, 36]. When 1-aminoanthracene was competed off the SIRT2—decanoyl-peptide complex, a dissociation constant of 4 μM was used as the $K_d$ value in the calculation [12].

## General cell culture methods

A549 human lung cancer cells were kept at 37˚C in the incubator with 5% $CO_2$. Cells were cultured in DMEM supplemented with 10% FBS, 100 U/ml penicillin, and 100 μg/ml streptomycin (full DMEM). To genetically knockout SIRT2 in A549 cells using CRISPR/Cas9, 250,000 cells were initially seeded into 6-well plates using Corning transfectagro media (catalog #40-300-CV), in which cells were cultured for 16 hours. Then, 1.5 μg of PX459 plasmid (addgene catalog #62988) [49] that encoded a SIRT2 guide RNA sequence targeting the protein coding region (CACCGCCGGCCTCTATGACAACCTA) was transfected into the cells. Transfection occurred by mixing the plasmid with 5 μl of Promega ViaFect reagent (catalog #E4981) for 5 minutes before transferring the mixture to 2 ml of fresh transfectagro, which was then applied to the cells. Cells were grown for 24 hours before the transfection reagents were removed. Cells were then washed with PBS and grown in full DMEM supplemented with 1μg/ml puromycin for three days, which killed ~90% of the cells. Clonal selection was then performed by seeding the remaining cells in 96-well plates with full DMEM using a limiting dilution of one cell per well. Colonies that formed in the 96-well plates were propagated, and after 5 passages, KO of SIRT2 expression was confirmed in various clones using western blot.

## SIRT2 Western blot

After washing cells with PBS, protein lysates were prepared from A549 cells or putative A549 SIRT2-KO cells using RIPA buffer with 1 mM DTT. The concentration of protein was determined using a Pierce detergent-compatible Bradford protein assay (Thermo Fisher catalog #23246). 20 μg of total protein was boiled in Laemmli buffer containing β-mercaptoethanol before SDS-PAGE on Bio-Rad precast 4–15% gels. After SDS-PAGE, proteins were transferred to a PVDF membrane, which was then blocked with 5% BSA in TBST for 1 hour at room temperature. After blocking, the membrane was incubated with primary antibody diluted 1:1000 in TBST overnight at 4˚C. Horseradish peroxidase (HRP)-conjugated secondary antibody was then incubated with the PVDF membrane at a dilution of 1:3000 in TBST for 1 hour at room temperature. After washing, SuperSignal West Pico plus chemiluminescent substrate (Thermo Fisher catalog #34579) was used to detect protein bands with an Azure Biosystems c400 imager. Antibodies used were a rabbit SIRT2 primary antibody (Cell Signaling Technology catalog #12650), an HRP-conjugated goat anti-rabbit secondary antibody (Invitrogen catalog #65–6120), a mouse alpha-tubulin primary antibody (Novus Biologicals catalog #NB120-11304), and an HRP-conjugated goat anti-mouse secondary antibody (Invitrogen catalog #62–6520). Unaltered, raw images for Western blots can be found in S11 Fig.

## Cell culture immunofluorescence for staining acetylated alpha-tubulin

100,000 A549 or A549 SIRT2-KO cells were seeded on a coverslip that was placed in the well of 6-well plate, and the cells were grown overnight in full DMEM. Cells were treated with **1** or SirReal2 for 6 hours (SirReal2 was purchased from Selleck Chemicals); the media from ligand-treated cells and control (DMSO only) cells contained a final concentration of 2% DMSO. The cells were then washed three times with PBS, fixed with 4% paraformaldehyde for 10 min, then washed again three times with PBS. Cells were incubated with 0.2% Triton X-100 for 15 minutes on a shaker followed by three more PBS washes. The cells were then blocked with 2% BSA in PBS for 1 hour at room temperature. Cells were washed twice with PBS, then incubated overnight on a rocker at 4˚C with a primary antibody targeting acetylated alpha-tubulin (Millipore Sigma catalog #T7451); this mouse monoclonal antibody was diluted 1:1000 in PBS containing 2% BSA. The cells were washed twice with PBS then incubated for 1 hour at room temperature with a secondary antibody diluted 1:2000 in PBS containing 2% BSA (Invitrogen

catalog #A-11001); this goat anti-mouse secondary antibody was conjugated to Alexa Fluor 488. After washing twice with PBS and once with autoclaved water, the coverslip with cells was removed from the 6-well plate and mounted to a glass slide using DAPI-containing mounting medium (Vector Laboratories catalog #H-2000). Slides were imaged using a Keyence BZ-X710 fluorescence microscope with the following filter settings: DAPI, excitation/emission was 300–400 nm/438-484 nm; Alexa Fluor 488, excitation/emission was 448 nm/500-550 nm. All images were obtained under identical conditions, and fluorescence intensities were quantified using Fiji/ImageJ without any manipulation to the images [50]. The fluorescence intensity from at least six cells per coverslip were randomly selected and measured targeting the cytosol (for measuring acetylated alpha-tubulin levels via Alexa Fluor 488) or the nucleus (for measuring nuclear DNA staining via DAPI). Relative fluorescence intensity was calculated as the ratio of Alexa Fluor 488/DAPI intensities. Statistical comparisons using one-way ANOVA and Dunnett's multiple comparison test were performed with GraphPad Prism.

## Cell viability assays

5,000 A549 cells were seeded in a 96-well plate in full DMEM. Cells were cultured overnight, then SirReal2 or **1** were added to the media at the final concentrations shown in Fig 4F and 4G for 72 hours. The total volume in each well was 150 μl, and the DMSO concentration was 0.7%. After this treatment period, 10 μl of MTT reagent was added to each well, then the cells were cultured for 4 hour before adding 100 μl of crystal dissolving solution for another 16 hour incubation (Cayman Chemical, catalog #10009365). The absorbance in each well was measured at 570 nm and was used to determine the cell viability by comparing the absorbance of drug treated cells to control (DMSO treated) cells.

## Supporting information

**S1 Fig. X-ray crystal structure of SIRT2 bound to FAM-myristoyl-H4K16 peptide (PDB code 8TGP).** The panels show surface or cartoon representations of SIRT2 colored dark gray from the same view, and the bound peptide was shown in sticks and colored by atom (carbon, green; nitrogen, blue; oxygen, red). Only five amino acid residues from the myristoylated peptide were visible in the crystal structure, as indicated.
(TIF)

**S2 Fig. Interaction of SIRT2 with myristoyl-H4K16 peptides using ITC.** (A) Binding isotherm and fit curve determining a $K_d$ of 13 nM for SIRT2's interaction with an unlabeled myristoyl-H4K16 peptide. (B) Binding isotherm and fit curve determining a $K_d$ of 25 nM for SIRT2's interaction with FAM-myristoyl-H4K16 peptide. ITC was performed in PBS using a Malvern Panalytical MicroCal PEAQ-ITC. The sample cell and syringe were set to 25°C. 20 μM SIRT2 was placed in the sample cell, and 200 μM peptide was placed in the ITC syringe. 19 injections of peptide were automatically applied to the sample cell in a 2 μl volume, except for the first injection (0.4 μl). A one site binding model was fit to the data by the software to calculate binding parameters.
(TIF)

**S3 Fig. The fluorescence of FAM-myristoyl-H4K16 peptide does not significantly change when bound to SIRT2 alone.** In this experiment, the fluorescence of 50 nM FAM-myristoyl-H4K16 peptide was measured in the presence or absence of 1 μM SIRT2 (excitation/emission wavelengths were 340 nm/520 nm, similar to HTRF assays). The fluorescence intensity differed by only 9.5% between the two conditions.
(TIFF)

**S4 Fig. 1% DMSO does not affect FRET signal in HTRF assays when SIRT2 binds FAM-myristoyl-H4K16 peptide.** In this experiment, assay conditions were 2.5 nM SIRT2, 0.2 nM terbium cryptate-labeled antibody, and 3 nM FAM-myristoyl-H4K16 peptide.
(TIFF)

**S5 Fig. 63 compounds, initially identified in the HTRF screen as inhibiting FAM-myristoyl-H4K16 binding to SIRT2 by >40%, were tested for their ability to displace the peptide from SIRT2 in full dose-response binding assays.**
(PDF)

**S6 Fig. Chemical structures of nine confirmed hit compounds from high-throughput screening.** The Compound IDs are also shown in Fig 2A of the main article.
(TIFF)

**S7 Fig.** Dose-response relationships showing the effects of 1 on (A) SIRT1 deacetylase activity and (B) SIRT6 demyristoylase activity.
(TIFF)

**S8 Fig. Dose-response relationships showing the effects of 2 and 15 on SIRT2 deacetylase and demyristoylase activities.** (A) **2** inhibited SIRT2 deacetylase activity with an $IC_{50}$ of 9 μM. (B) **2** had no effect on SIRT2 demyristoylase activity. (C) **15** had no effect on SIRT2 deacetylase activity. (D) **15** inhibited SIRT2 demyristoylase activity with an $IC_{50}$ of 117 μM. (E) **15** did not efficiently displace Cy3-myristoyl-H4K16 peptide from SIRT2 at concentrations as high as 100 μM. The dotted line indicates the background fluorescence (peptide only) where 100% inhibition of binding would occur. (F) **15** did not efficiently displace 1-aminoanthracene from SIRT2 at concentrations as high as 100 μM. The dotted line indicates the background fluorescence (1-aminoanthracene only) where 100% inhibition of binding would occur.
(TIFF)

**S9 Fig. Interaction of 4, 5, and 6 with SIRT2 as measured with a 1-aminoanthracene binding competition assay.** For all assays, the SIRT2 concentration was constant at 4 μM and the 1-aminoanthracene concentration was 100 nM. The dotted line in the panels shows the fluorescence of 1-aminoanthracene alone in the absence of SIRT2, which was the baseline for 100% inhibition of 1-aminoanthracene binding. (A) **4** competed 1-aminoanthracene from SIRT2 with an $IC_{50}$ of 128 μM, which was used to calculate a $K_d$ of 116 μM for the interaction of **4** with SIRT2. (B) **5** competed 1-aminoanthracene from SIRT2 with an $IC_{50}$ of 130 μM, which was used to calculate a $K_d$ of 117 μM for the interaction of **5** with SIRT2. (C) **6** competed 1-aminoanthracene from SIRT2 with an $IC_{50}$ of 81 μM, which was used to calculate a $K_d$ of 73 μM for the interaction of **6** with SIRT2.
(TIFF)

**S10 Fig. 4-Bromo-2-iodophenol (2) does not covalently react with SIRT2.** (A) MALDI-MS spectra of 10 μM SIRT2 after incubation in PBS with 1% DMSO at 37˚C for 30 minutes. (B) MALDI-MS spectra of 10 μM SIRT2 after incubation with 100 μM of **2** in PBS with 1% DMSO at 37˚C for 30 minutes. The predicted monoisotopic mass for this SIRT2 protein was 36552 Da, and the predicted average mass was 36576 Da; thus, the main peaks represented +1 ions in the isotopic cluster. Notably, the observed mass of the protein does not change after incubation with the ligand. The intact protein spectra were acquired with a Bruker microflex instrument in linear, positive ion mode using sinapinic acid as the matrix. Sinapinic acid was dissolved in 50% acetonitrile/50% water with 0.1% trifluoroacetic acid, which was also used to dilute protein prior to spotting on the MALDI plate.
(TIFF)

**S11 Fig. Original, unaltered Western blot images.** Per journal policy, we provide (A) the original Western blot for SIRT2 and (B) the original Western blot for alpha-tubulin, both of which are shown in Fig 4C. For both blots, the experiment contained a third lane that was cropped out of the main text figure and is not relevant for the reported results.
(TIF)

**S1 Table. Chemical characteristics of investigated compounds (mass, volume, and polarity).**
(PDF)

## Acknowledgments

The authors thank Dr. David Bi for collecting preliminary data on SIRT2-ligand interactions.

## Author Contributions

**Conceptualization:** Jie Yang, Joel Cassel, Brian P. Weiser.

**Funding acquisition:** Brian P. Weiser.

**Investigation:** Jie Yang, Joel Cassel, Brian C. Boyle, Daniel Oppong.

**Methodology:** Jie Yang, Joel Cassel, Brian C. Boyle, Daniel Oppong, Young-Hoon Ahn, Brian P. Weiser.

**Supervision:** Brian P. Weiser.

**Writing – original draft:** Jie Yang, Brian P. Weiser.

**Writing – review & editing:** Jie Yang, Joel Cassel, Brian C. Boyle, Daniel Oppong, Young-Hoon Ahn, Brian P. Weiser.

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
