## [Decision Letter · Decision Letter 0]

26 Apr 2024

PONE-D-24-07409A Homogeneous Time-Resolved Fluorescence Screen to Identify SIRT2 Deacetylase and Defatty-Acylase InhibitorsPLOS ONE

Dear Dr. Weiser,

Thank you for submitting your manuscript to PLOS ONE. After careful consideration, we feel that it has merit but does not fully meet PLOS ONE’s publication criteria as it currently stands. As you can see from all three reviewers, they raised major concerns that warrant major revision. However, I believe most of these are addressable. Therefore, we invite you to submit a revised version of the manuscript that addresses the points raised during the review process.

We look forward to receiving your revised manuscript.

Kind regards,

Michael Massiah

Academic Editor

PLOS ONE

Journal Requirements:

Reviewers' comments:

Reviewer's Responses to Questions

**Comments to the Author**

1. Is the manuscript technically sound, and do the data support the conclusions?

Reviewer #1: Partly

Reviewer #2: Partly

2. Has the statistical analysis been performed appropriately and rigorously? 

Reviewer #1: Yes

Reviewer #2: Yes

3. Have the authors made all data underlying the findings in their manuscript fully available?

Reviewer #1: Yes

Reviewer #2: No

4. Is the manuscript presented in an intelligible fashion and written in standard English?

Reviewer #1: Yes

Reviewer #2: No

5. Review Comments to the Author

Reviewer #1: I have carefully read the manuscript of Jie Yang et al. and felt that the research was potentially important and interesting. In this study, the authors developed a new time-resolved FRET-based SIRT2 binding assay system based on their previously reported fluorescence peptide assay (ref. 32). By means of this system, they performed chemical screening using 9600 compound library and found 9 true hit compounds. The hit 1 showed moderate SIRT2 deacetylase and defatty-acylase inhibitions, and interestingly, two fragments of hit 1, namely 2 and 15, showed SIRT2 deacetylase and defatty-acylase selective inhibitions, respectively. From this perspective, I recommend to accept this paper for publication in PLoS One after some revisions.

My concerns:

1. In Figure 1C and D, the value of relative HTRF signal of high-conc. competitor was different. What does this difference mean?

2. In the screening, the authors obtained 63 compounds as hit compounds, but most of them did not show reproducible SIRT2 inhibition. I think that this result clearly shows a limitation of HTRF screening system they developed. I would like the authors to discuss the reasons why so many false-positive compounds were generated in the revised manuscript.

3. The authors described: “The unique selectivity of sirtuins for different acyl modifications on proteins creates challenges and opportunities for drug discovery. Apart from creating isoform-specific inhibitors, there is interest in modulating select deacylase activities of sirtuins which likely have different roles in the cell.” and I agree with this statement, but I am wandering how about the isoform-specificity of hit compound 1. The author had better show the specificity among SIRT family (SIRT1-7).

4. The author described: “For 1, the IC50 values for inhibiting deacetylase and de-decanoylase activities (7-8 �M) were five-fold more potent than the IC50 values for inhibiting de-dodecanoylase and demyristoylase activities (37-39 �M) (Figures 3B-3E).” I think that the authors should discuss the reasons and/or speculations.

5. The authors noted: “As discussed elsewhere [37,38], SIRT2 inhibition has varying effects on the growth rates of different cell types.” And it is true that many of SIRT2 inhibitors exhibited cytotoxicity via c-Myc, however hit 1 did not. What does this result mean?

6. The authors’ group previously reported that propofol allostericaly inhibits SIRT2 deaetylase activity. Does this report suggest that compound 15 and 1 bind to another allosteric site of SIRT2 that only affect defatty-acylase activity? I think that the author stated the details in the revised manuscript.

Reviewer #2: Generally, the assay is of interest to a wide community and could be publilshed in the journal.

The abstract should make it clearer that also inhibitors of just deacetylation might/will show an inhibition. SirReal2 should be tested for a deacetylase inhibitor and e.g. TM as a commercially available inhibitor of both activities in addition to the ascorbate which might should unspecific redox reactivity. The structures of all hits should be shown. The compound shows a high potential to be a Pan assay interference substans (PAINS). This should be discussed and checked with other sirtuins and maybe unrelated enzymes. Acetyl tubulin is a highly unspecific readout which should be discussed, this is not a a direct target engament assay and the ascorbate concentration is very high. For this, a CETSA like in the articles of Olsen should be performed. Vogelmann et al present compound 12 as low µM inhibitor, this should be cited. cpd 12 has actually a Ki of 13 nM according to Zessin et al.

6. PLOS authors have the option to publish the peer review history of their article (what does this mean?). If published, this will include your full peer review and any attached files.

Reviewer #1: No

Reviewer #2: **Yes: **Manfred Jung

---

## [Author Response · Author response to Decision Letter 0]

17 May 2024

PONE-D-24-07409

A Homogeneous Time-Resolved Fluorescence Screen to Identify SIRT2 Deacetylase and Defatty-Acylase Inhibitors

PLOS ONE

Comments to the Author

Authors’ Response: We sincerely thank the Reviewers for spending their time evaluating our manuscript. We have addressed the comments below in our Responses and have made appropriate changes in our manuscript (highlighted yellow in the markup copy). We believe that this Review process genuinely improved our manuscript and hope that the revised manuscript is found suitable for publication. 

Reviewer #1: I have carefully read the manuscript of Jie Yang et al. and felt that the research was potentially important and interesting. In this study, the authors developed a new time-resolved FRET-based SIRT2 binding assay system based on their previously reported fluorescence peptide assay (ref. 32). By means of this system, they performed chemical screening using 9600 compound library and found 9 true hit compounds. The hit 1 showed moderate SIRT2 deacetylase and defatty-acylase inhibitions, and interestingly, two fragments of hit 1, namely 2 and 15, showed SIRT2 deacetylase and defatty-acylase selective inhibitions, respectively. From this perspective, I recommend to accept this paper for publication in PLoS One after some revisions.

My concerns:

1. In Figure 1C and D, the value of relative HTRF signal of high-conc. competitor was different. What does this difference mean?

Authors’ Response: In the revision, we have elaborated on this observation in the Results and Discussion where we present Figures 1C, 1D, and (new) Figure 1E. High concentrations of unlabeled myristoyl-peptide very efficiently and competitively displaced the fluorescent myristoyl-peptide from SIRT2 by ~90% (Figure 1C). However ascorbyl palmitate only partially reduced the HTRF signal by ~40% (Figure 1D). This may indicate that ascorbyl palmitate is only a partial inhibitor of SIRT2 and not a true competitive inhibitor. This is important to note in the context of new Figure 1E where SirReal2 was also found to only partially displace the fluorescent myristoyl-peptide from SIRT2. This is notable because SirReal2, despite its potency as a deacetylase inhibitor, is not a competitive inhibitor of SIRT2’s demyristoylase activity (Spiegelman, 2018, ChemMedChem). 

2. In the screening, the authors obtained 63 compounds as hit compounds, but most of them did not show reproducible SIRT2 inhibition. I think that this result clearly shows a limitation of HTRF screening system they developed. I would like the authors to discuss the reasons why so many false-positive compounds were generated in the revised manuscript.

Authors’ Response: We believe that we addressed this comment in the revision with the following text, italicized below for emphasis:

These 63 compounds were then examined with the same HTRF assay in full dose-response experiments to confirm binding (S5 Fig). Only 17% of the initial hits (11 of the 63 compounds) were confirmed to reduce FAM-myristoyl-H4K16 peptide binding by >40% (S5 Fig); additional plate replicates may be needed to reduce the false positive rate in future screens. In the full dose-response assays, we found that 9 compounds displaced FAM-myristoyl-H4K16 peptide from SIRT2 with IC50 values less than 10 uM (Fig 2A) (see S6 Fig for the chemical structures of all 9 compounds).

3. The authors described: “The unique selectivity of sirtuins for different acyl modifications on proteins creates challenges and opportunities for drug discovery. Apart from creating isoform-specific inhibitors, there is interest in modulating select deacylase activities of sirtuins which likely have different roles in the cell.” and I agree with this statement, but I am wandering how about the isoform-specificity of hit compound 1. The author had better show the specificity among SIRT family (SIRT1-7).

Authors’ Response: We performed additional experiments for this revision testing the ability of compound 1 to inhibit isoforms SIRT1 and SIRT6 in addition to SIRT2. Compound 1 inhibited SIRT1 deacetylase activity with an IC50 of 32 uM (compared to an IC50 of 7 uM for inhibiting SIRT2 deacetylase activity), and compound 1 had no effect on SIRT6 demyristoylase activity. This text is now in the Results and Discussion where we present SIRT2 activity assays in Figure 3, and the SIRT1 and SIRT6 data is shown in new Figure S7. This should provide some initial context for the ligand’s selectivity; we hope to examine other sirtuins in the future when we are better equipped with the full panel of enzymes, along with their unique substrates, and additional analogs of compound 1. 

4. The author described: “For 1, the IC50 values for inhibiting deacetylase and de-decanoylase activities (7-8 uM) were five-fold more potent than the IC50 values for inhibiting de-dodecanoylase and demyristoylase activities (37-39 uM) (Figures 3B-3E).” I think that the authors should discuss the reasons and/or speculations.

Authors’ Response: We have commented on this observation in the revision:

“For 1, the IC50 values for inhibiting deacetylase and de-decanoylase activities (7-8 uM) were five-fold more potent than the IC50 values for inhibiting de-dodecanoylase and demyristoylase activities (37-39 uM) (Figs 3B-3E). This was interesting because we consider decanoylation to be a long fatty acyl modification with relatively high affinity for SIRT2 compared to acetylation [12,33], and 1 preferentially inhibited deacylase activity at a very specific acyl chain length.”

We hesitate to speculate further why this occurs without additional evidence. Some ligands with much higher affinity for SIRT2 selectively inhibit deacylase activities for reasons that are not always clear, even with structural information about their interactions.

5. The authors noted: “As discussed elsewhere [37,38], SIRT2 inhibition has varying effects on the growth rates of different cell types.” And it is true that many of SIRT2 inhibitors exhibited cytotoxicity via c-Myc, however hit 1 did not. What does this result mean?

Authors’ Response: We elaborated on this result in the revision with the following text:

 “Finally, as discussed elsewhere [37,38], SIRT2 regulates diverse processes in cancer cells depending on the tissue type and tumor stage, causing the enzyme to be called both a tumor suppressor and oncogene, and its inhibition has varying effects on cellular growth rates. In our case, we found that the SIRT2 inhibitors SirReal2 and 1 had no effect on the viability of A549 cells as measured with an MTT assay (Figs 4F and 4G).”

6. The authors’ group previously reported that propofol allostericaly inhibits SIRT2 deaetylase activity. Does this report suggest that compound 15 and 1 bind to another allosteric site of SIRT2 that only affect defatty-acylase activity? I think that the author stated the details in the revised manuscript.

Authors’ Response: This is an interesting question that we appreciate from the Reviewer. Our initial report was indeed that propofol is a weak allosteric inhibitor of SIRT2 deacetylase activity (Weiser, 2015, JBC). However that work was conducted at a time when SIRT2’s defatty-acylase activity was not fully understood. As we learned more about SIRT2’s enzymology and the structural basis for its demyristoylase activity, we expanded our findings to conclude that while propofol is an allosteric deacetylase inhibitor, it is a competitive demyristoylase inhibitor (Bi…Weiser, 2020, Biochemistry). This occurs because propofol occupies SIRT2’s hydrophobic tunnel at a site where myristoyl modifications, but not acetyl modifications, interact. 

Our data in this manuscript (Figure 6) certainly indicates that the fragment compound 2, and probably the same fragment of compound 1, binds in the propofol site in the hydrophobic tunnel. Whether fragment compound 15 would orient in the hydrophobic tunnel or in an adjacent allosteric site is not yet clear. In the revision, we included a discussion about this in a short paragraph at the end of the Results and Discussion section, as follows:

“The experiment in Fig 6 positions the binding site of fragment 2 at the end of the hydrophobic tunnel near residues that interact with 3 and 1-aminoanthracene, including Tyr139, Phe190, and Leu206 [12,41]. This is an allosteric site relative to a bound acetyl-lysine substrate, but a competitive site relative to a bound myristoyl-lysine. In the context of 1 binding to SIRT2, whether fragment 15 would orient inside the hydrophobic tunnel or in an adjacent allosteric site to inhibit demyristoylase activity is not yet clear.”

Reviewer #2: Generally, the assay is of interest to a wide community and could be publilshed in the journal.

The abstract should make it clearer that also inhibitors of just deacetylation might/will show an inhibition. 

Authors’ Response: In light of our data in Figures 1E, 2B, and 2C, we now included the following sentence in the Abstract:

“The high-throughput screen also detected multiple deacetylase-specific SIRT2 inhibitors.”

(see also the next Authors’ Response)

SirReal2 should be tested for a deacetylase inhibitor and e.g. TM as a commercially available inhibitor of both activities in addition to the ascorbate which might should unspecific redox reactivity. 

Authors’ Response: In the revision, we tested SirReal2 binding to SIRT2 in HTRF format (new Figure 1E). We report that SirReal2 partially inhibited the interaction of SIRT2 with the fluorescein-labeled myristoyl peptide by 70% with an IC50 of 2 uM. This finding contributed to the statement that we now included in the Abstract that the assay can also detect SIRT2 interactions with deacetylase-specific inhibitors.

We find that analyzing TM binding to SIRT2 in the HTRF assays may not provide a significant advantage as a positive control compared to ascorbyl palmitate. First, TM requires a preincubation step with NAD+ in order to effectively inhibit the demyristoylase activity of the enzyme (Spiegelman, 2018, ChemMedChem), whereas ascorbyl palmitate does not. Whether the fluorescence of NAD+ interferes with the HTRF assay has not been tested. Additionally, we have measured direct interactions of ascorbyl palmitate with SIRT2 using multiple biophysical techniques including SPR (Hong…Weiser, 2021, ChemMedChem), which suggests that ascorbyl palmitate binding to SIRT2 is not related to potential redox reactivity. Thus we believe that ascorbyl palmitate serves as a reasonable positive control as a known defatty-acylase/deacetylase inhibitor in the development of the HTRF assay as a test of whether it displaces the FAM-labeled peptide. 

The structures of all hits should be shown. 

Authors’ Response: These are now shown in Figure S6.

The compound shows a high potential to be a Pan assay interference substans (PAINS). This should be discussed and checked with other sirtuins and maybe unrelated enzymes. 

Authors’ Response: In the revision, we tested the hit compound 1 against SIRT1 and SIRT6. We report that compound 1 inhibited SIRT1 deacetylase activity with an IC50 of 32 uM (compared to an IC50 of 7 uM for inhibiting SIRT2 deacetylase activity), and compound 1 had no effect on SIRT6 demyristoylase activity. (See also point 3 in our response to Reviewer 1). Thus, the ligand does not simply interfere non-specifically with any sirtuin or enzyme reaction.

Acetyl tubulin is a highly unspecific readout which should be discussed, this is not a a direct target engament assay and the ascorbate concentration is very high. For this, a CETSA like in the articles of Olsen should be performed. 

Authors’ Response: We have addressed this important point in the revision. The following paragraph was re-written for the Results and Discussion (italics are included here for emphasis):

“We then examined whether 1 could inhibit SIRT2 in cells. Inhibition of SIRT2’s deacetylase activity in cells can result in elevated levels of acetylated alpha-tubulin, which is an in vivo substrate of SIRT2 [16,18,19,26,36]. As measured with immunofluorescence, treatment of cells with 100 uM of 1 significantly increased the level of acetylated alpha-tubulin (Figs 4A and 4B). We performed two control experiments to further substantiate that the change in acetylated alpha-tubulin resulted from SIRT2 inhibition by 1 because this readout indirectly reports on SIRT2 activity in cells, as opposed to directly measuring engagement of the ligand with the enzyme [37]. First, we treated cells with 25 uM of the SIRT2 deacetylase inhibitor SirReal2 [26], which increased acetylated alpha-tubulin to similar levels as 1 (Figs 4A and 4B). Secondly, we generated A549 SIRT2-KO cells (Fig 4C), which also had an elevated level of acetylated alpha-tubulin that was unchanged by the SIRT2 inhibitors (Figs 4D and 4E).” 

Vogelmann et al present compound 12 as low µM inhibitor, this should be cited. cpd 12 has actually a Ki of 13 nM according to Zessin et al.

Authors’ Response: We thank the Reviewer for pointing out these important references that were mistakenly omitted. We have amended the Introduction accordingly to include these compounds.

“Newly identified molecules of the “SirReal” class have shown the most promise as SIRT2 demyristoylase inhibitors with potencies in the mid nM to low uM range[23,24]. However, other small molecules that inhibit SIRT2’s demyristoylase activity have lower selectivity or potency and include ascorbyl palmitate (IC50 = ~8 to 23 uM) [25], 1-aminoanthracene (IC50 = 21 uM) [12], “compound C” (IC50 = 44 uM) [26], and suramin (IC50 = 95 uM) [27]. It may be difficult to assess the therapeutic value of targeting different deacylase activities of SIRT2 without additional classes of defatty-acylase inhibitors.”

---

## [Decision Letter · Decision Letter 1]

22 May 2024

A Homogeneous Time-Resolved Fluorescence Screen to Identify SIRT2 Deacetylase and Defatty-Acylase Inhibitors

PONE-D-24-07409R1

Dear Dr. Weiser,

We’re pleased to inform you that your manuscript has been judged scientifically suitable for publication and will be formally accepted for publication once it meets all outstanding technical requirements.

Kind regards,

Michael Massiah

Academic Editor

PLOS ONE

Additional Editor Comments (optional):

Reviewers' comments:

Reviewer's Responses to Questions

**Comments to the Author**

1. If the authors have adequately addressed your comments raised in a previous round of review and you feel that this manuscript is now acceptable for publication, you may indicate that here to bypass the “Comments to the Author” section, enter your conflict of interest statement in the “Confidential to Editor” section, and submit your "Accept" recommendation.

Reviewer #1: All comments have been addressed

2. Is the manuscript technically sound, and do the data support the conclusions?

Reviewer #1: Yes

3. Has the statistical analysis been performed appropriately and rigorously? 

Reviewer #1: Yes

4. Have the authors made all data underlying the findings in their manuscript fully available?

Reviewer #1: Yes

5. Is the manuscript presented in an intelligible fashion and written in standard English?

Reviewer #1: Yes

6. Review Comments to the Author

Reviewer #1: The authors adequately addressed all my concerns. I have no further comment.

7. PLOS authors have the option to publish the peer review history of their article (what does this mean?). If published, this will include your full peer review and any attached files.

Reviewer #1: No

---

## [Editor Report · Acceptance letter]

13 Jun 2024

PONE-D-24-07409R1 

PLOS ONE

Dear Dr. Weiser, 

I'm pleased to inform you that your manuscript has been deemed suitable for publication in PLOS ONE. Congratulations! Your manuscript is now being handed over to our production team.

Kind regards, 

on behalf of

Dr. Michael Massiah 

Academic Editor

PLOS ONE